# Contribution of primary care expansion to Sustainable Development Goal 3 for health: a microsimulation of the 15 largest cities in Brazil

Sanjay Basu ,[1] Thomas Hone ,[2] Daniel Villela,[3] Valeria Saraceni,[4] Anete Trajman,[5] Betina Durovni,[6] Christopher Millett,[2] Davide Rasella[7]

¹Research and Development, Waymark, San Francisco, California, USA
²Public Health Policy Evaluation Unit, Imperial College London, London, UK
³Program of Scientific Computing, Oswaldo Cruz Foundation, Rio de Janeiro, Brazil
⁴Secretaria Municipal de Saude do Rio de Janiero, Rio de Janeiro, Brazil
⁵Centro de Estudos Estrategicos, Oswaldo Cruz Foundation, Rio de Janeiro, Brazil
⁶Federal University of Rio de Janeiro, Rio de Janeiro, Brazil
⁷ISGlobal, Barcelona, Spain

**Correspondence to**
Dr Sanjay Basu;
Sanjayb493@gmail.com

## ABSTRACT

**Objectives** As middle-income countries strive to achieve the Sustainable Development Goals (SDGs), it remains unclear to what degree expanding primary care coverage can help achieve those goals and reduce within-country inequalities in mortality. Our objective was to estimate the potential impact of primary care expansion on cause-specific mortality in the 15 largest Brazilian cities.

**Design** Microsimulation model.

**Setting** 15 largest cities by population size in Brazil.

**Participants** Simulated populations.

**Interventions** We performed survival analysis to estimate HRs of death by cause and by demographic group, from a national administrative database linked to the Estratégia de Saúde da Família (Family Health Strategy, FHS) electronic health and death records among 1.2 million residents of Rio de Janeiro (2010–2016). We incorporated the HRs into a microsimulation to estimate the impact of changing primary care coverage in the 15 largest cities by population size in Brazil.

**Primary and secondary outcome measures** Crude and age-standardised mortality by cause, infant mortality and under-5 mortality.

**Results** Increased FHS coverage would be expected to reduce inequalities in mortality among cities (from 2.8 to 2.4 deaths per 1000 between the highest-mortality and lowest-mortality city, given a 40 percentage point increase in coverage), between welfare recipients and non-recipients (from 1.3 to 1.0 deaths per 1,000), and among race/ethnic groups (between Black and White Brazilians from 1.0 to 0.8 deaths per 1,000). Even a 40 percentage point increase in coverage, however, would be insufficient to reach SDG targets alone, as it would be expected to reduce premature mortality from non-communicable diseases by 20% (vs the target of 33%), and communicable diseases by 15% (vs 100%).

**Conclusions** FHS primary care coverage may be critically beneficial to reducing within-country health inequalities, but reaching SDG targets will likely require coordination between primary care and other sectors.

## INTRODUCTION

Increasing access to primary care has been linked to reduced mortality at both the individual and population levels.[1–7] Primary

### Strengths and limitations of this study

► This study quantified the degree to which expansion or contraction of Brazil's largest primary care programme would be expected to help achieve the Sustainable Development Goals (SDGs), and the implications of programme expansion or contraction on inequalities.

► The findings suggest that primary care coverage may be critically beneficial to reducing within-country health inequalities, but reaching the SDG targets would be unlikely without additional resources and efforts from other sectors.

► The study helps direct emphasis towards coordination between primary care and other sectors, including efforts to address the wider socioeconomic determinants of health.

► The principal limitations arise from being based on a simulation model that cannot account for unobserved confounders.

► Additionally, the infant mortality and under-5 mortality outcomes were based on HRs from the literature rather than from detailed individual-level data, due to limitations in data availability from the primary datasets we used.

care expansion in low-income and middle-income countries remains a major strategy for reducing mortality, and for achieving the United Nations Sustainable Development Goals (SDGs).[8] Indeed, primary care expansion is listed by the United Nations as a key intervention to achieve the SDG mortality reduction targets, which include reducing premature mortality from non-communicable diseases by one-third, ending deaths from communicable diseases, and reducing under-5 mortality to <25 per 1000 live births by the year 2030.[9]

The degree to which improving primary care access can further contribute to reductions in mortality remains unclear. Better evidence is needed to drive policy-making,

given that global mortality targets are often out of sync with local health system planning and budgeting activities and insufficiently tailored to local contexts.[10 11] Divestments by national governments from the primary care sector have partially been justified by the lack of clear evidence that primary care can be expected to achieve targets such as the SDGs.[12–14] Local governments often vary in their ability and willingness to fund primary care independently of federal support. Hence, particularly in countries with decentralised decision-making, it is vital to estimate expected declines in mortality from primary care investments, and conversely whether targets can be better designed with local and regional baseline conditions and primary care effectiveness in mind.[15]

In this study, we estimate the potential impact of primary care expansion on mortality by cause of death and by age in the 15 largest Brazilian cities. We use a simulation model incorporating microlevel demographic, health and effect-size data from Brazil's Estratégia de Saúde da Família (Family Health Strategy, FHS) primary care programme. The FHS is Brazil's main strategy for achieving universal healthcare, and is based on primary care delivery through multidisciplinary care teams colocated in clinics; mobile community healthcare worker teams trained to extend clinic reach; and evidence-based training, protocol management and record-keeping systems including a unique administrative dataset for tracking health outcomes among individual FHS users and non-users.[1 2] Each team includes a physician, nurses and community healthcare workers responsible for delivering maternal and child healthcare, curative care, health promotion and prevention, chronic disease management, home visits and referrals to a catchment population of approximately 1000 families (~3450 individuals). In 2020, 5462 local government municipalities (out of 5565 in Brazil) had these health teams, covering 133.9 million individuals (63.7% of the population).[16] Since 1996, municipal governments have been responsible for financing and delivery of primary care in Brazil. In the context of national government cuts to healthcare budgets, local governments have varied in the degree to which they augment primary care investments.[12 13]

Our prior work has shown that expanded FHS coverage in Brazil has been associated with reductions in both non-communicable and communicable disease mortality, infant and under-5 mortality, as well as health disparities among race/ethnic and urban/rural groups.[1 2 7 12] Here, we simulate the fifteen largest Brazilian cities with detailed demographic and health data—incorporating information on variations in FHS primary care coverage, and estimate relationships among coverage to mortality risk at an individual level by cause to project how further expansion or contraction of the FHS programme may affect crude and age-standardised mortality by cause, infant mortality and under-5 mortality, and to compare such mortality rate variations to international targets for mortality. We focus on cities because they have an average FHS coverage of 35%—far below the national coverage—yet constitute the

largest proportion of under-served favela (slum) populations who are the intended primary target populations for FHS.[1]

## METHODS
### Model structure
We designed and implemented a microsimulation model,[17] which simulates individual people in each of the largest fifteen Brazilian cities, their demographics, their risk of specific causes of death conditional on their location and demographics, and the estimated change in their risk of death given expansion or contraction of the FHS primary care programme, over the period 2020–2030. A microsimulation is a model that can be envisioned as a large table, where each row is an individual and each column is a characteristic (ie, location, demographic, health status) of the individual. Within the microsimulation, probabilities of death by cause by year (derived from annual mortality rate by cause) are conditioned on the location and demographics of the individual, as well as whether they have access to the FHS primary care programme. We adjusted those probabilities of death to simulate changes in FHS coverage, per the data sources detailed below.

### Relationship between FHS primary care coverage and mortality outcomes
FHS effects estimates were obtained from a previous retrospective study on the association between FHS usage and mortality by age, socioeconomic status and cause of death for 1.2 million residents of Rio de Janeiro (2010–2016).[18] Specifically, a survival analysis was carried out using the linked *Cadastro Único* national administrative database,[19] FHS electronic health records and mortality records. Flexible parametric survival analysis models were used to estimate HRs for each ICD-10-CM cause of death and cause of death disease group (eg, Neoplasms are ICD-10 codes C00 through D48[20] by sex, race, and if the family receives benefits from the national conditional cash transfers programme Bolsa Familia or not. The models were inverse probability treatment weighted to adjust for the probability of FHS participation, and regression adjusted for age, highest level of education, disability, unemployment, household per capita income decile, number of family members per bedroom, family size, number of children in family, household flooring, household piped water access, quintiles of household expenditure on medicines, quintile of per capita household expenditure on food, formal labour employment and if the individual has been hospitalised before FHS use. Results are displayed in figure 1. By contrast with cause-specific adult mortality, HRs for infant mortality and under-5 mortality were obtained from a prior systematic review of literature, as routine administrative data were not available in disaggregated form for children.[21]

### Simulation of changing FHS primary care coverage levels
We used data from the Brazilian Institute of Geography and Statistics on fifteen Brazilian cities (including Rio de

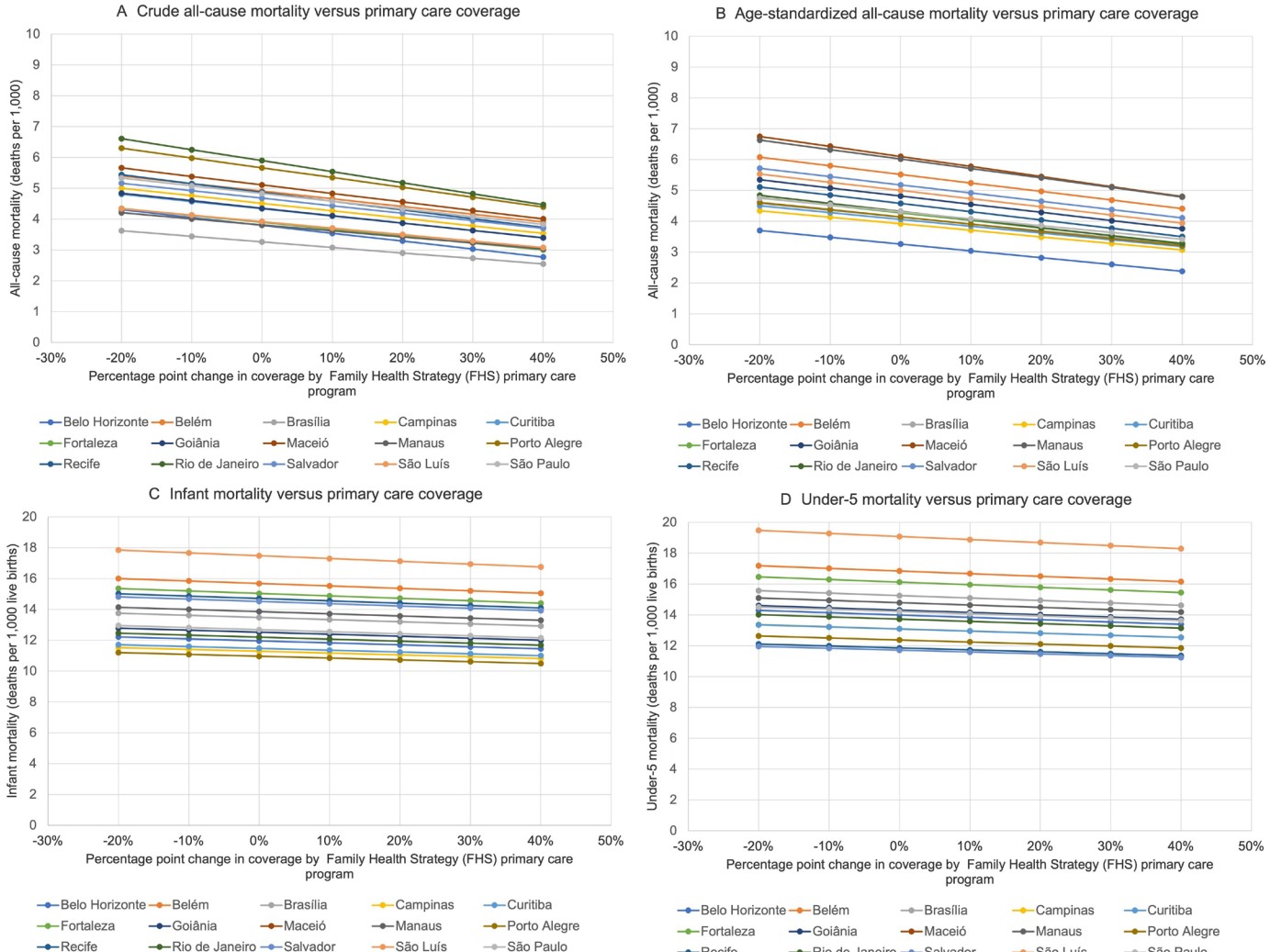

**Figure 1** Projected variations in (A) all-cause crude mortality, (B) all-cause age-standardised mortality, (C) infant mortality and (D) under-5 mortality given different levels of FHS programme primary care coverage. See table 1 for current coverage levels corresponding to a 0% change on the x axis. See 95% CIs in figure 2.

Janeiro) to generate a simulated population of each city.[22] We generated the simulated representative population based on the demographic characteristics, FHS enrolment probability and mortality risks of each population, as itemised in table 1. We then varied the FHS coverage level in each city, and applied the HRs estimated from the Rio de Janeiro survival analysis (described above) to estimate the impact of changing FHS coverage on mortality in all of the simulated cities, through the method specified below.

We specifically estimated the impact of changing coverage on mortality in two steps. First, we calculated the base mortality probability for each simulated person in the absence of FHS primary care, using the following formula:

$m_b \times HR \times p + m_b \times (1 - p) = m_c$ (Eq. 1), where $m_b$ is the base mortality probability for each cause of death in each city without FHS (to be estimated); HR is the HR for that cause of death for FHS users versus non-users obtained from the Rio de Janeiro survival analysis; p is the latest (2016 observed) FHS-covered proportion of the

population in the simulated city, and $m_c$ is the latest (2016 observed) mortality probability in that city. We used 2016 because it was the last year of data available for the Rio de Janeiro analysis.[18]

Second, after calculating the base mortality probability for each cause of death in each city, we then estimated the new mortality probability from equation 1, conditional on a simulated FHS coverage level, to re-estimate the deaths by cause under different FHS coverage levels. We varied the FHS coverage level from 20 percentage points below the current observed coverage level to 40 percentage points above the current observed coverage level in each city (up to a maximum of 100 percent). The choice of 40 percentage points assumed that the FHS expansion policy would have been coordinated at the federal level, with an effort to progress from 60% to 100% coverage by 2030.

The primary outcome was change in all-cause mortality. Secondary outcomes included changes in cause-specific and all-cause mortality subgrouped by race/ethnicity, and whether the family receives Bolsa Familia benefits or not,

**Table 1** Demographics, primary care coverage, and mortality among the fifteen largest Brazilian cities

| City | Age, less than 15 years, % | Age, greater than 64 years, % | Female, % | Race, % White | Race, % Black | Ethnicity, % Pardo | Income, % below poverty line | Education, % graduated secondary school (age 18–20) | FHS primary care coverage, % | Mortality, all cause, crude (per 1,000) | Mortality, all cause, age adjusted (per 1,000) | Infant mortality (per 1000 live births) | Under-5 mortality (per 1000 live births) |
|---|---|---|---|---|---|---|---|---|---|---|---|---|---|
| Belo Horizonte | 29.9 | 4.7 | 52.7 | 40.5 | 12.6 | 46.5 | 5.6 | 47.5 | 76.5 | 5.76 | 4.95 | 13.0 | 15.2 |
| Belém | 24.8 | 5.5 | 52.1 | 21.2 | 9.1 | 69.3 | 14.9 | 37.1 | 23.5 | 5.49 | 6.17 | 16.1 | 17.2 |
| Brasília | 23.7 | 5.0 | 52.2 | 40.0 | 10.6 | 48.3 | 4.9 | 53.5 | 37.8 | 3.93 | 4.98 | 14.0 | 15.8 |
| Campinas | 19.3 | 5.8 | 51.8 | 57.2 | 8.9 | 30.8 | 3.2 | 53.2 | 44.5 | 5.60 | 4.86 | 11.8 | 13.7 |
| Curitiba | 20.0 | 4.9 | 52.3 | 74.3 | 3.8 | 58.5 | 1.7 | 57.8 | 36.4 | 5.19 | 4.86 | 11.9 | 13.6 |
| Fortaleza | 22.6 | 4.9 | 53.2 | 32.3 | 5.8 | 61.4 | 12.1 | 45.4 | 45.9 | 4.93 | 5.40 | 15.8 | 16.9 |
| Goiânia | 20.8 | 4.2 | 52.3 | 43.2 | 6.4 | 49.8 | 3.1 | 57.0 | 44.3 | 5.42 | 5.99 | 13.1 | 15.0 |
| Maceió | 25.0 | 4.4 | 53.2 | 27.1 | 5.6 | 66.7 | 15.6 | 42.6 | 29.0 | 5.91 | 7.04 | 22.0 | 24.0 |
| Manaus | 28.2 | 3.2 | 51.2 | 20.1 | 2.3 | 77.0 | 12.9 | 38.8 | 27.0 | 4.34 | 6.84 | 14.2 | 15.2 |
| Porto Alegre | 18.8 | 6.4 | 53.6 | 78.0 | 11.4 | 10.2 | 3.8 | 48.2 | 55.0 | 7.42 | 5.43 | 11.6 | 13.1 |
| Recife | 20.9 | 5.5 | 53.8 | 36.9 | 9.1 | 52.7 | 13.2 | 46.7 | 54.8 | 6.43 | 6.05 | 15.6 | 12.5 |
| Rio de Janeiro | 19.4 | 7.6 | 53.2 | 48.5 | 11.6 | 39.2 | 5.0 | 45.9 | 62.9 | 8.14 | 5.97 | 13.0 | 14.6 |
| Salvador | 20.7 | 4.3 | 53.3 | 14.8 | 38.0 | 46.8 | 11.4 | 41.8 | 26.7 | 5.32 | 5.90 | 14.9 | 12.0 |
| São Luís | 23.7 | 3.8 | 53.2 | 19.9 | 15.5 | 63.9 | 13.8 | 53.1 | 34.4 | 4.64 | 5.91 | 18.1 | 19.8 |
| São Paulo | 20.8 | 5.5 | 52.6 | 57.9 | 8.7 | 30.4 | 4.3 | 50.5 | 35.4 | 5.71 | 5.09 | 13.2 | 14.7 |

Estratégia de Saúde da Família (FHS) primary care programme (2016).[19]
FHS, Family Health Strategy.

considering the differential HRs of death by cause and by these characteristics, as described above.

## Uncertainty analyses

At each level of simulated FHS coverage, we repeatedly simulated each of the fifteen cities' populations a total of 10 000 times each. During each of these simulations, we sampled with replacement from random normal distributions constructed around the mean and 95% CIs around the mean of the HRs of FHS primary care enrolment on each cause of death (supporting materials figure 1, supporting materials table 1), to generate uncertainty intervals around our outcome estimates. All analyses were performed in *R*.

## Patient and public involvement

No patient involved.

## RESULTS
## Population characteristics

Population demographics, FHS primary care programme coverage rates and mortality rate estimates by city are provided in table 1. Notable demographics included the population aged less than 15 years (which varied between 18.8% to 29.9%), the populations aged above age 64 years (between 3.2% and 7.7%), race/ethnicity (between 14.8% and 78.0% identifying as White, between 2.3% and 38.0% black, and 10.2%–77.0% Pardo). City poverty rates varied from 1.7% to 15.6%. Age-standardised all-cause mortality rates varied from 4.9 to 7.0 per 1000 population,

infant mortality varied from 11.8 to 22.0 per 1000 live births, and under-5 mortality from 12.0 to 24.0 per 1000 live births. (Hence, all cities had under-5 mortality rates below 25 per 1000 live births, meeting SDG target 3.2).

## Projected mortality variations with changes in primary care coverage

In our simulations, we varied the FHS coverage level from 20 percentage points below the current observed coverage level to 40 percentage points above the current observed coverage level in each city (up to a maximum of 100 percent). Projected variations in crude and age-standardised all-cause mortality, as well as in infant mortality and under-5 mortality, are illustrated in figures 1 and 2. Variations in cause-specific age-standardised mortality are itemised in table 2. Uncertainty estimates (95% CIs) around all estimates are provided in figure 2 and table 2. In general, increases in FHS coverage were associated with reductions in predicted mortality.

As shown in figure 1A,B, increases in FHS coverage would be expected to contribute to reductions in inequalities between cities in both crude-cause and all-cause mortality. The associations between FHS coverage and reductions in mortality were greatest in cities with the highest rates of baseline mortality. For example, a 20 percentage point decline in coverage would be expected to increase crude mortality in Rio de Janeiro by 12% (from a current level of 5.9–6.6 per 1000 (95% CI 6.2 to 7.0)), and increase crude mortality in Sao Paulo by 10% (from 4.8 to 5.3 per 1000 (95% CI 5.2 to 5.4)). By contrast, as shown in

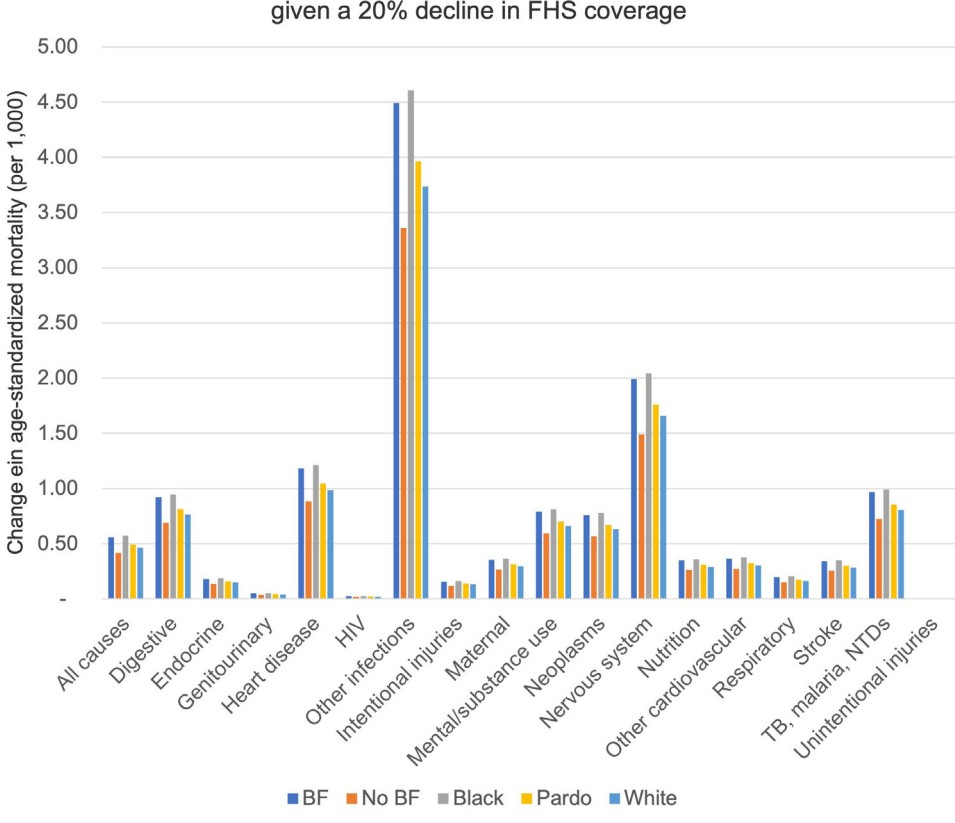

### A Change in age-standardized mortality by subgroup, given a 20% decline in FHS coverage

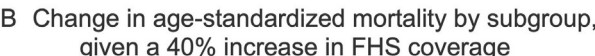

■ BF  ■ No BF  ■ Black  ■ Pardo  ■ White

### B Change in age-standardized mortality by subgroup, given a 40% increase in FHS coverage

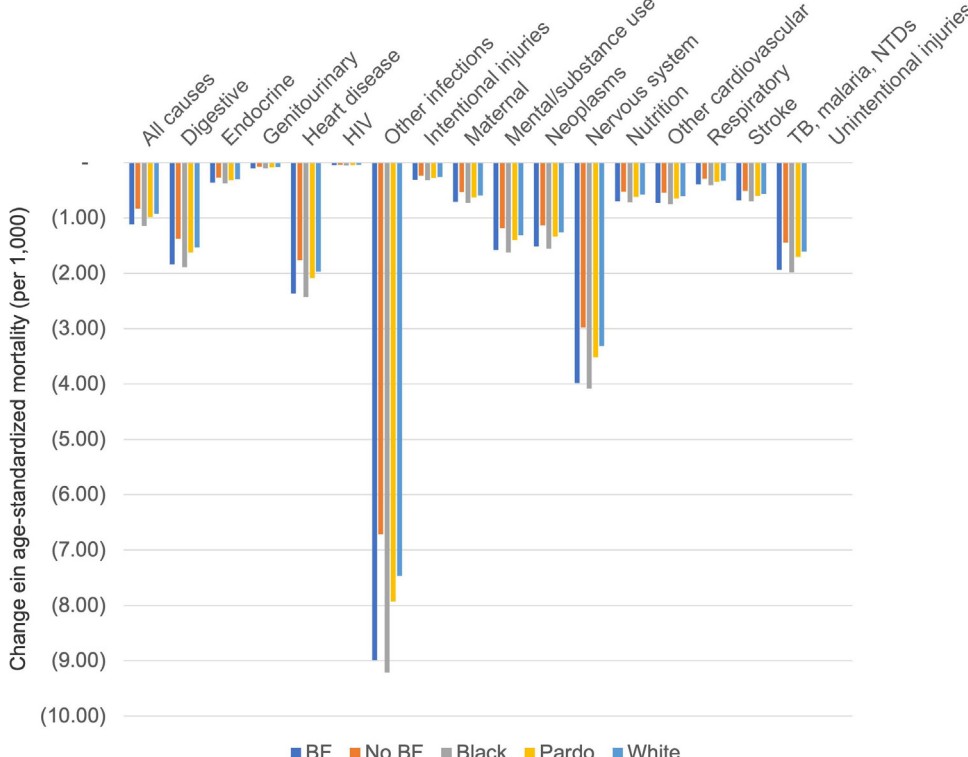

■ BF  ■ No BF  ■ Black  ■ Pardo  ■ White

**Figure 2** Projected changes in all-cause and cause-specific age-standardised mortality given (A) a 20 percentage point decline or (B) a 40 percentage point increase in Family Health Strategy (FHS) programme primary care coverage from the baseline levels indicated in table 1. BF, participation in the Bolsa familia programme.

**Table 2** Relative impact on cause-specific mortality given changes in the percentage point coverage in the FHS primary care strategy (Estratégia de Saúde da Família). TB: tuberculosis. NTDs: Neglected Tropical Diseases.

| | Ratio of mortality by cause, compared with current mortality rate (at 0%) | | | | | | |
|---|---|---|---|---|---|---|---|
| **Percentage point change in FHS coverage** | **−20%** | **−10%** | **0%** | **10%** | **20%** | **30%** | **40%** |
| All causes | 1.11 | 1.05 | 1.00 | 0.95 | 0.89 | 0.84 | 0.78 |
| Infections (excluding HIV, TB, malaria, NTDs) | 1.11 | 1.05 | 1.00 | 0.95 | 0.89 | 0.84 | 0.79 |
| HIV | 1.08 | 1.04 | 1.00 | 0.96 | 0.92 | 0.88 | 0.84 |
| TB, malaria, NTDs | 1.03 | 1.02 | 1.00 | 0.98 | 0.97 | 0.95 | 0.94 |
| Respiratory | 1.10 | 1.05 | 1.00 | 0.95 | 0.90 | 0.85 | 0.80 |
| Nutrition | 1.08 | 1.04 | 1.00 | 0.96 | 0.92 | 0.87 | 0.83 |
| Neoplasms | 1.11 | 1.05 | 1.00 | 0.95 | 0.89 | 0.84 | 0.78 |
| Nervous system | 1.06 | 1.03 | 1.00 | 0.97 | 0.94 | 0.92 | 0.89 |
| Endocrine | 1.11 | 1.05 | 1.00 | 0.95 | 0.89 | 0.84 | 0.79 |
| Mental/substance use | 1.10 | 1.05 | 1.00 | 0.95 | 0.90 | 0.84 | 0.79 |
| Stroke | 1.14 | 1.07 | 1.00 | 0.93 | 0.86 | 0.80 | 0.73 |
| Heart disease | 1.11 | 1.05 | 1.00 | 0.95 | 0.89 | 0.84 | 0.79 |
| Other cardiovascular | 1.13 | 1.06 | 1.00 | 0.94 | 0.87 | 0.81 | 0.74 |
| Digestive | 1.10 | 1.05 | 1.00 | 0.95 | 0.90 | 0.84 | 0.79 |
| Genitourinary | 1.10 | 1.05 | 1.00 | 0.95 | 0.90 | 0.86 | 0.81 |
| Unintentional injuries | 1.13 | 1.07 | 1.00 | 0.93 | 0.87 | 0.80 | 0.74 |
| Intentional injuries | 1.16 | 1.08 | 1.00 | 0.92 | 0.84 | 0.75 | 0.67 |
| Maternal | 1.01 | 1.00 | 1.00 | 1.00 | 0.99 | 0.99 | 0.99 |

The cells show the ratio of mortality by cause under different levels of FHS coverage, compared with the current mortality rate (at 0% change in FHS coverage), the reference column. 95% CIs in table 2.
FHS, Family Health Strategy.

figure 1, when primary care coverage rates increased, the model observed less inequality among cities, with diminishing returns on further reductions in mortality among those already with low mortality rates. For example, a 20 percentage point increase in coverage would be expected to reduce crude mortality by 21% in Sao Paulo (from 4.8 to 3.8 per 1000 (95% CI 3.3 to 4.4)) and by 24% in Rio de Janeiro (from 5.9 to 4.5 per 1000 (95% CI 3.4 to 5.5)). A 40 percentage point increase in coverage would be expected to reduce overall between-city all-cause mortality (differences in mortality between the highest and lowest mortality city) from 2.8 to 2.4 deaths per 1000 (95% CI 2.3 to 2.6). The same pattern was observed with infant mortality (figure 1C) and under-5 mortality (figure 1D).

As shown in table 2, the specific causes of mortality that were most sensitive to the changes in FHS primary care coverage were chronic non-communicable disease deaths, including cardiovascular disease deaths and deaths from injury (n.b., many FHS clinics are often the first point of service for injuries). According to the model, a 20 percentage point reduction in coverage would be expected to raise deaths from unintentional and intentional injuries by 13% (95% CI 9% to 17%) and 16% (95% CI 13% to 20%), respectively, and from heart disease and stroke from 11% (95% CI 7% to 14%) and 14% (95% CI 10% to 17%), respectively. Conversely,

the smallest changes in cause-specific mortality were for deaths from nervous system diseases, tuberculosis, malaria, neglected tropical diseases, and maternity. For the SDG of reducing by one-third premature mortality (death prior to age 70 years) from non-communicable diseases, the results implied that even an increase in FHS primary care coverage by 40 percentage points would still insufficient by itself to reach the target, as premature mortality from non-communicable diseases was only estimated to fall by 20% (95% CI 7% to 34%). For the SDG of ending deaths from communicable diseases, the model results implied that an increase in FHS primary care coverage by 40 percentage points would be expected to reduce mortality from communicable diseases by 15% (95% CI 1% to 29%).

### Subgroup analyses
The model was used to estimate changes in mortality across groups defined by race/ethnicity and whether the family receives Bolsa Familia benefits, considering the differential HRs of death by cause and by these characteristics (table 1). Across all causes of death, FHS primary care coverage disproportionately benefited Black and Pardo groups, and those on Bolsa Familia benefits. For each percentage point decline (or increase) in primary care FHS coverage, the absolute increase (or decrease) in mortality was 1.3 times higher for Bolsa Familia families

than for non-recipient families, 1.2 times higher for Black and 1.1 times higher for Pardo families than for White families (figure 2). A 40 percentage point increase in coverage would be expected to reduce inequality in mortality between welfare recipients and non-recipients from 1.3 to 1.0 deaths per 1000 (95% CI 0.8 to 1.2), inequality between Black and White Brazilians from 1.0 to 0.8 deaths per 1000 (95% CI 0.6 to 0.9), and inequality between Pardo and White Brazilians from 0.3 to 0.2 per 1000 (95% CI 0.1 to 0.3).

## DISCUSSION

We simulated the impacts of changes in FHS coverage on mortality in the 15 largest Brazilian cities using detailed demographic and health data, and estimated relationships at an individual level by cause of death, controlling for major covariates. We found reductions in FHS coverage would be expected to lead to higher mortality and exacerbate inequalities among cities. Additionally, marginalised groups including those receiving Bolsa Familia and those of minority race/ethnic groups would be expected to disproportionately benefit from increasing FHS coverage, and so increased FHS coverage would also be expected to drive reductions in inequalities within cities. We estimated that an increase in FHS primary care coverage by 40 percentage points would still be insufficient on its own, however, to reach SDG target 3.4, as it would only reduce premature mortality from non-communicable diseases by 20% (vs the SDG target of 33%). Additionally, our results implied that even an increase in FHS primary care coverage by 40 percentage points would be expected to reduce mortality from communicable diseases by 15% but still be far from the SDG of ending deaths from communicable diseases. By contrast, Brazilian cities had already reached SDG target 3.2 of reducing under-5 mortality to less than 25 per 1000 live births.[23]

The study findings suggest that, in the context of a health system with decentralised decision making to over 5000 municipal governments, geographical inequality in mortality in Brazil will potentially be greatly affected by future federal financing and support of the FHS. Investment in primary care may be beneficial for local achievement of SDGs for both non-communicable and communicable diseases and reduce geographical inequality in these outcomes. Nonetheless, localities would need to engage with non-health sectors to achieve an elimination of health inequalities and should consider setting bolder local targets for accelerating under-5 mortality rates beyond those prescribed in the SDGs.

Our results are particularly important considering the current economic crisis due to the COVID-19 pandemic in Brazil, which is already responsible for a dramatic increase in poverty and unemployment and will have long-term effects on the most vulnerable groups of the population.[24] Our findings are consistent with previous studies which have shown a synergistic impact of FHS with Bolsa Familia (the country's social welfare programme)

on child mortality,[25 26] and on premature mortality during the past Brazilian economic recession.[27] Other simulation studies, performed at the aggregate-level, have also indicated how a combined expansion of FHS and Bolsa Familia during periods of economic crisis are able to reduce the number of childhood deaths.[12] While our study is focused on Brazilian cities, recent literature has also demonstrated the effectiveness of interventions to expand the primary care coverage to rural areas, as the Mais Medicos Programme, on the reduction of premature mortality,[28] and have shown how weakening of such intervention during periods of economic austerity could be responsible for a large number of avoidable deaths.[14 29]

Brazil is one of the few low-income and middle-income countries with a universal healthcare system, the Unified Health System (SUS), based on one of the world's largest primary healthcare programmes, the FHS. The expansion of the SUS and FHS during the last 30 years was responsible for large reductions on mortality and health inequalities,[30] but is currently under threat by aggressive and long-term fiscal austerity measures that could undermine its consolidation and even reduce its dimension and effectiveness,[13 14] so robust evidence on the impact of its components—including FHS, are urgently needed.

Our study has several strengths and limitations. Our simulation model used estimates of FHS impact on adult mortality based on a unique, individual-level dataset with a rich set of covariates including key socioeconomic and health variables that relate to mortality; nevertheless, factors other than those we have controlled for may be additionally important to consider and may serve as unmeasured factors influencing our results, such as prior family history of disease. Additionally, this dataset was restricted to the city of Rio de Janeiro where FHS impacts may differ from those in the other major cities included in our study. Indeed, the FHS in Rio has major investments in clinics and equipment, a residency programme, and higher salaries for doctors. Our simulation of infant mortality and under-5 mortality was based on HRs from the literature rather than from detailed individual-level data, due to limitations in data availability from the primary datasets we used. This increases the possibility that inequalities across key socioeconomic groups would be underestimated by using a group measure, and hence the overall impact of the primary care programme may be underestimated.

In future research, we plan to examine how rural populations respond differently to urban populations to FHS primary care coverage expansion and identify factors that may help enhance the effectiveness of FHS in reducing mortality across Brazil. Our work also suggests future research should include more comparative analysis of the health impacts delivered by different models of primary care in Latin America and worldwide. In the meantime, our results suggest that increasing primary care coverage may be critically beneficial to achieving SDG targets including reducing within-country inequalities between

geographical areas, income and race/ethnic groups in Brazil.

**Contributors** SB serves as guarantor of the article. CM and DR conceived of the study. VS, AT and BD collected data. SB, TH and DR performed the analysis. SB wrote the draft. All authors contributed to editing of the manuscript.

**Funding** This work was supported by a grant from the Health Systems Research Initiative with funding from the Foreign, Commonwealth and Development Office, the UK Medical Research Council and Wellcome, in collaboration with the UK Economic and Social Research Council (grant no. MR/P014593/1).

**Competing interests** None declared.

**Patient consent for publication** Not applicable.

**Ethics approval** Approval for this study was obtained from the Brazilian National Commission for Ethics in Research (Comissão Nacional de Ética em Pesquisa (CONEP); number 2.689.528) and Imperial College London's Ethical Committee.

**Provenance and peer review** Not commissioned; externally peer reviewed.

**Data availability statement** All data relevant to the study are included in the article or uploaded as online supplemental information. Technical appendix, statistical code, and dataset available from https://github.com/sanjaybasu concurrent with publication.

**ORCID iDs**
Sanjay Basu http://orcid.org/0000-0002-0599-6332
Thomas Hone http://orcid.org/0000-0003-0703-6973

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
