## [Reviewer comments · BMJ Open]

ARTICLE DETAILS

TITLE (PROVISIONAL)	The contribution of primary care expansion to Sustainable Development Goal Three for health: A microsimulation of the fifteen largest cities in Brazil
AUTHORS	Basu, Sanjay; Hone, Thomas; Villela, Daniel; Saraceni, Valeria; Trajman, Anete; Durovni, Betina; Millett, Christopher; Rasella, Davide

VERSION 1 – REVIEW

REVIEWER	Tomasi, Elaine Universidade Federal de Pelotas, Medicina Social
REVIEW RETURNED	24-Mar-2021

GENERAL COMMENTS	First of all, I congratulate the authors for the excellent paper. Abstract - OK Introduction - clearly poses the problem and situates it in the face of evidence and gaps ... well-justified study With an innovative methodology, it analyzes and projects results from large databases in a rigorous and consistent manner. Among its limitations, it points out that “simulation model that cannot account for unobserved confounders”. Could the authors provide some examples of these confounders? The authors also point out that the weakness that “... the infant mortality and under-5 mortality outcomes were based on hazard ratios from the literature rather than from detailed individual-level data, due to limitations in data availability from the primary datasets we used”. It would be useful to answer: Could this limitation have printed what or what effect (s) on the measurements obtained? Since it is an important component used as outcomes, the authors could provide more details of these data from the literature, as not all readers will have access to the referred article. Innovative model for analyzing large databases, using what is available; great merit in developing and testing models; What are the recommendations for replicating the model for other outcomes and exposures? The article brings encouraging results by highlighting the benefits of public policies in the social area, with emphasis on health. The subgroup analysis carried out for race / color and for Bolsa Família more explicitly explains the effect of policies on more vulnerable
---

	groups. This is more important in times of austerity in the country's economic policy, which perversely affects social policies, making the living and health conditions of the vast majority of the population worse. Table 1 could inform the population of each city; It is not clear the color scheme adopted in table 2 and what the numbers 0 and 1 represent above the mortality ratios.
--	---

REVIEWER	Suresan, Vinay Hitkarini Dental College and Hospital
REVIEW RETURNED	26-Mar-2021

GENERAL COMMENTS	A Research article with new insights in role played by the Primary health care setup to achieve SDG's.
--

REVIEWER	Chen, Yingxi National Institutes of Health
REVIEW RETURNED	07-Jul-2021

GENERAL COMMENTS	Thank you, editor, for inviting me to review this manuscript. Basu et al conducted a microsimulation analysis of to estimate the potential impact of primary care expansion on cause-specific mortality in the 15 largest Brazilian cities. The study was well-designed and presented. I only had several questions/comments:  1. There are a couple of typos; please check. 2. I am not sure exactly how the cause of mortality identified, specifically what codes? 3. "The infant mortality and under-5 mortality outcomes were based on hazard ratios from the literature rather than from detailed individual-level data." How this could affect the interpretation of the study results? Also, because mortality and cause-specific mortality estimates were from de-detailed individual data, would the analyses be comparable across the three mortality groups?
--

REVIEWER	Costa-Font, Joan London School of Economics and Political Science
REVIEW RETURNED	19-Jul-2021

GENERAL COMMENTS	The paper examines an interesting and important question about the effect of FHS on mortality and inequality. However, the methods used are more up to the challenge. One would need a full-fledged survival analysis to answer this question so that it disentangles differences relevant controls that could bias the estimates, or even worst that could explain both the treatment (coverage for FHS) and the outcome. The question on inequalities, and specifically ethnic inequalities in important, but it only plots estimates by ethnicity which could be driven by other covariates. The methods to simulate the effect of FHS is not explained. I thought this needs a lot more detail. The text of this paper need a lot of work, a native author would be recommended. There are numerous typos all over.
---

VERSION 1 – AUTHOR RESPONSE

Reviewer suggestion	Revision made	Location of revision
Reviewer 1		
First of all, I congratulate the authors for the excellent paper. Abstract - OK Introduction - clearly poses the problem and situates it in the face of evidence and gaps ... well-justified study With an innovative methodology, it analyzes and projects results from large databases in a rigorous and consistent manner.	Thank you.	-
Among its limitations, it points out that “simulation model that cannot account for unobserved confounders”. Could the authors provide some examples of these confounders?	We now clarify and provide examples of which confounders might be unaccounted for by the survival analysis and microsimulation model.	Discussion section, p. 14
The authors also point out that the weakness that “... the infant mortality and under-5 mortality outcomes were based on hazard ratios from the literature rather than from detailed individual-level data, due to limitations in data availability from the primary datasets we used”. It would be useful to answer: Could this limitation have printed what or what effect (s) on the	We now clarify what effect we would anticipate from the unavailability of individual-level data for the infant and under-5 group, specifically the possibility that inequalities across key socioeconomic groups would be underestimated by using a group measure, and hence the overall impact of the primary care program may be underestimated.	Discussion section, p. 14

measurements obtained? Since it is an important component used as outcomes, the authors could provide more details of these data from the literature, as not all readers will have access to the referred article.		
Innovative model for analyzing large databases, using what is available; great merit in developing and testing models;	Thank you.	-
What are the recommendations for replicating the model for other outcomes and exposures?	We now note that the model code is shared such that others can replicate the model and extend it to other outcomes and exposures.	Data sharing statement.
The article brings encouraging results by highlighting the benefits of public policies in the social area, with emphasis on health. The subgroup analysis carried out for race / color and for Bolsa Família more explicitly explains the effect of policies on more vulnerable groups. This is more important in times of austerity in the country's economic policy, which perversely affects social policies, making the living and health conditions of the vast majority of the population worse.	Thank you, we now further clarify and concur the importance of the work in the context of austerity.	Discussion section
Table 1 could inform the population of each city;	Thank you.	-

It is not clear the color scheme adopted in table 2 and what the numbers 0 and 1 represent above the mortality ratios.	We have removed the color scheme to reduce confusion, and have now adjusted the caption to clarify the interpretations of the numbers above the mortality ratios.	Table 2
Reviewer 2		
A Research article with new insights in role played by the Primary health care setup to achieve SDG's.	Thank you.	-
Reviewer 3		
Basu et al conducted a microsimulation analysis of to estimate the potential impact of primary care expansion on cause-specific mortality in the 15 largest Brazilian cities. The study was well-designed and presented. I only had several questions/comments: 1. There are a couple of typos; please check.	We have revised the manuscript extensively to improve language choice and phrasing throughout, and to correct typos.	Throughout.
I am not sure exactly how the cause of mortality identified, specifically what codes?	We now clarify that we used the ICD-10 code set to group mortality codes. We additionally upload and share the code that was used to classify the codes into groups per the ICD standard grouping rubric.	Methods p. 7
“The infant mortality and under-5 mortality outcomes were based on hazard ratios from the literature rather than from detailed individual-level data.” How this could affect the interpretation of the study results? Also, because mortality and cause-specific mortality	We now clarify that inequalities across key socioeconomic groups would be underestimated by using a group measure, and hence the overall impact of the primary care program may be underestimated. We specify additionally in the Discussion that for this reason, the adult analyses may be considered	Discussion, p. 14

estimates were from de-detailed individual data, would the analyses be comparable across the three mortality groups?	less likely to underestimate effect than the infant or under-5 estimates.	
Reviewer 4		
The paper examines an interesting and important question about the effect of FHS on mortality and inequality. However, the methods used are more up to the challenge. One would need a full-fledged survival analysis to answer this question so that it disentangles differences relevant controls that could bias the estimates, or even worst that could explain both the treatment (coverage for FHS) and the outcome.	We now clarify that we did indeed perform a full survival analysis, which we further detail in the revised Methods section and provide the complete results of in the Appendix. We specify the control variables included in that analysis, including the inverse probability weighted approach we used to reduce the risk of bias.	Methods p. 7, Appendix
The question on inequalities, and specifically ethnic inequalities in important, but it only plots estimates by ethnicity which could be driven by other covariates.	We clarify as above that the estimates control for the other available covariates now further clarified in the Methods section.	Methods p. 7
The methods to simulate the effect of FHS is not explained. I thought this needs a lot more detail.	See above regarding the survival analysis.	Methods p. 7
The text of this paper need a lot of work, a native author would be recommended. There are numerous typos all over.	We have revised the manuscript extensively to improve language choice and phrasing throughout, and to correct typos.	Throughout.

VERSION 2 – REVIEW

REVIEWER	Tomasi, Elaine Universidade Federal de Pelotas, Medicina Social
REVIEW RETURNED	27-Aug-2021

GENERAL COMMENTS	The authors complied with the requests and the article is ready for publication.
--

REVIEWER	Chen, Yingxi National Institutes of Health
REVIEW RETURNED	10-Sep-2021

GENERAL COMMENTS	The authors have adequately addressed my comments. I don't have any further comments.
---